# Microstructural Analysis and Tribological Behavior of AMDRY 1371 (Mo–NiCrFeBSiC) Atmospheric Plasma Spray Deposited Thin Coatings

**Cornelia Cîrlan Paleu [1], Corneliu Munteanu [1,*], Bogdan Istrate [1], Shubrajit Bhaumik [2], Petrică Vizureanu [3,*], Mădălina Simona Bălțatu [3] and Viorel Paleu [1,*]**

[1] Department of Mechanical Engineering, Mechatronics and Robotics, Faculty of Mechanical Engineering, Gheorghe Asachi Technical University of Iaşi, 63 D. Mangeron Blvd., 700050 Iaşi, Romania; cpaleu@yahoo.com (C.C.P.); bogdan_istrate1@yahoo.com (B.I.)

[2] Tribology and Surface Interaction Research Laboratory, Department of Mechanical Engineering, SRM Institute of Science and Technology, Kattankulathur 603203, India; shubrajb@srmist.edu.in

[3] Department of Technologies and Equipments for Materials Processing, Faculty of Materials Science and Engineering, Gheorghe Asachi Technical University of Iaşi, Blvd. No. 51, Mangeron, 700050 Iasi, Romania; cercel.msimona@yahoo.com

* Correspondence: corneliu.munteanu@academic.tuiasi.ro (C.M.); peviz@tuiasi.ro (P.V.); vpaleu@tuiasi.ro (V.P.); Tel.: +40-744-793-984 (P.V.)

**Abstract:** Water treatment plants include a set of pumping stations, and their mechanical components experience various wear modes. In order to combat wear, the mechanical components of the pumps are coated with various types of wear resistant coatings. In this research, AMDRY 1371 (Mo–NiCrFeBSiC) coatings were deposited with the atmospheric plasma spray (APS) method on parallelepipedal steel samples manufactured from a worn sleeve of a multistage vertical irrigation pump. In order to find an optimum thickness of AMDRY 1371 coatings, the samples were coated with five, seven and nine passes (counted as return passes of the APS gun). Mechanical properties of the coating (microhardness and Young's modulus) were determined by micro-indentation tests. An AMSLER tribometer was used to investigate the wear resistance and wear modes of the coated samples in dry conditions. A mean coefficient of friction (CoF) of around 0.3 was found for all the samples, but its evolution during the one hour of the test and also the final wear volumes and wear rates depended on the thickness of the coating. To estimate the roughness of the surfaces and the wear volumes, measurements were carried out on a Taylor Hobson profilometer. In order to understand the nature and evolution of wear of coatings of various thicknesses, the unworn and worn surfaces of the coated samples were analyzed by scanning electron microscopy (SEM), energy-dispersive X-ray spectroscopy (EDS), and X-ray diffraction (XRD). The wear modes of the coatings were studied, emphasizing the coating removal process for each sample. According to our results, for each dry friction application, there is an optimum value of the thickness of the coating, depending on the running conditions.

**Keywords:** coating; AMDRY 1371; Mo–NiCrFeBSiC; Mo–NiCrBSi; wear; friction; atmospheric plasma spray; scanning electron microscopy; energy-dispersive X-ray spectroscopy; pumps

---

## 1. Introduction

Pumps are the heart of any water treatment plant. Their mechanical components, such as impellers, rolling bearings, seals, bushes, and sleeves, suffer severe damage due to various wear mechanisms: corrosion, abrasion, adhesion, erosion, cavitation, pitting etc. One of the ways to combat such kind of combined wear is by coating the surfaces with wear resistant powders. These powders can be

deposited on substrate material by different coating techniques, such as atmospheric plasma spray (APS), cold spray (CS), high velocity oxy fuel (HVOF) etc. Azarmi et al. [1] optimized the deposition parameters for the APS process of 625 Ni-based superalloy, emphasizing that the oxidation index and the melting index incorporate all the parameters that were found to be significant in the statistical analyses and correlate well with the measured oxide content and porosity in the coatings. Rico et al. [2] reported results on nanostructured and conventional coatings of $Al_2O_3$–13% $TiO_2$ deposited by APS on SAE-42 steel, and found that the wear rate of the nanostructured material was lower than that achieved by the conventional coating. Deshpande et al. [3] compared the tribological behavior of $TiO_2$ APS coatings with low carbon steel APS coatings and substrate steel material. The experiments were carried out in a boundary lubrication regime in the presence of molybdenum dialkyl-dithiocarbamate (MoDTC) and a reduction in friction of 15–20% in the case of $TiO_2$ APS coatings was reported. Marquer et al. [4] investigated the wear properties of CuNiIn and CoCrAlYSiBN between titanium alloys ($Ti_6Al_4V$) in dry conditions. A good coating integrity was seen in the case of both coatings. They also reported that the frictional properties are more dependent on the contact pressure rather than on sliding velocity. The coatings did not have influence on the frictional coefficient but both of them reduced the wear rate to an extent. Xu et al. [5] incorporated SiC particles in Ni-60 powders and coated stainless steel (SS) 304 and found that 5% SiC incorporated Ni-60 powders exhibited an enhancement of tribological properties as compared to Ni-60 without SiC particles, both for dry and water lubrication conditions. Zhang et al. [6] deposited AT13/Mo coatings on SS 304 using APS and investigated their wear properties against $Si_2N_4$ balls using a ball-on-disc tribometer. They observed that molybdenum (Mo) plays an important role in controlling the frictional properties of AT13 powders, these being enhanced with the addition of Mo, and the friction was less in the AT13/Mo coated SS as compared to AT13 coated SS surfaces. Xiao et al. [7] studied the atmospheric thermal sprayed high entropy FeCoNiCrSiAl$_x$ alloy and reported a decrease in wear due to the presence of aluminum. Hashemi et al. [8] deposited $Cr_2O_3$-20YSZ (CZ) and $Cr_2O_3$-20YSZ-10SiC (CZS) using APS composite coatings on 304 L stainless steel and observed a wear intensification of the composite coatings as compared to $Cr_2O_3$ coating due to the phase transformation of zirconia. The corrosion resistance of the composite coatings was also better in case of the composite coatings as compared to the $Cr_2O_3$ coating due to the blocking of pores as a result of the tribofilms from the zirconia phase transformation. The wear resistance properties of $Cr_2O_3$ can be improved if it is doped with other different additives—$CeO_2$ and $Nb_2O_5$ [9]. The addition of these additives resulted in higher hardness and denser coatings as compared to $Cr_2O_3$ coatings without additives. $CeO_2$ exhibited better wear resistance than $Nb_2O_5$. Bhosale and Rathod [10] deposited WC–$Cr_3C_2$–Ni on SS 316 using atmospheric plasma and HVOF and reported an enhancement of anti-wear properties of both coated samples and enhanced anti-wear characteristics compared with uncoated specimens. Bolelli et al. [11] studied the abrasive wear resistance of HVOF deposited Fe–Cr–Ni–Si–B–C (Colferoloy) coatings, but the results were not so encouraging when compared against Ni-based alloys and electroplated chromium. Anyway, blending Fe–Cr–Ni–Si–B–C composite coatings with 20 wt.% and 40 wt.% of a WC–12 wt.% Co powder and spraying this composite powder with HVOF and high velocity air-fuel (HVAF) processes improved the abrasive wear resistance of Fe-based coatings [12]. Flame thermal spray duplex system composed of NiCrFeBSi plus Ni-P electroless nickel extended the corrosion resistance of NiCrFeBSi, with the Ni–P surface coating sealing the porosity of the coating [13]. Miguel et al. [14] realized a comparison between the wear modes of the NiCrBSi coating deposited by APS, HVOF, and sprayed/fused processes, with the wear behavior of the APS deposited coating being found the worst.

Molybdenum-based coatings provide low friction and scuffing resistance particularly at high temperature, being used in aerospace, automotive, pulp and paper, and plastic processing applications. Results on air plasma sprayed Mo coatings were reported by Sampath and Wayne [15], recommending the addition of $Mo_2C$ to NiCrFeBSi powder for improved kinetic friction properties. The $MoSi_2$–Mo composite coating possesses better wear resistance than a simple Mo coating, the wear mechanisms of both coatings being local plastic deformation, delamination, oxidation, and adhesion [16]. Various types

of Mo–NiCrFeBSiC coatings were proposed by Sampath and Vanderpool [17] and the values of the dry coefficient of friction (CoF) obtained by tests on a ball-on-disc tribometer were very high, the lowest CoF being 0.66 for 10 N normal load and 0.5 m/s sliding speed (440-C ball sliding against a coated disc). The results presented in [17] were not so encouraging, with high values of the dry friction coefficient being reported. Yegunov et al. [18] investigated the features of the coaxial laser gas powder surfacing (CGPS) of the AMDRY 1371 powder alloy (also known as Mo+NiCrBSi alloy) and reported low porosity and low fracture susceptibility of the deposited coatings, with better results at low speed and high power consumption, but no tribological study was carried out on the AMDRY 1371 coating. Preliminary research of Niranatlumpong and Koiprasert [19] reported that the effect of added Mo on tribological properties of NiCrBSi plasma sprayed coatings is beneficial for wear resistance with a small ratio (25 wt.%) of added Mo. Zhang et al. [20] added 5 wt.%–30 wt.% Mo to NiCrBSi and carried out dry and oil lubricated reciprocating friction tests. It was revealed that 30% of Mo in NiCrBSi provides the best tribological behavior in both dry and lubricated tests. Nevertheless, the lowest value of dry friction CoF was 0.6, which is very high, but the wear rate was drastically reduced by about 96% when compared to pure NiCrBSi. Dilawary et al. [21] added only 10 wt.% of Mo to NiCrBSi and reported that the wear resistance of plasma transfer arc (PTA) deposited hardfacing increased at both room and high temperatures (300–700 °C) due to the formation of Mo oxides. The measured CoF during ball-on-disc tests was between 0.4 and 0.9, the highest CoF being obtained for NiCrBSi + 10 wt.% at room temperature. Liu et al. [22] investigated the effect of heat treatment at 300, 500, and 700 °C of atmospheric plasma sprayed NiCrBSi coatings on their microstructure, phase composition, microhardness and tribological performance. Inter-splat oxidation of the coatings took place at high temperatures, reducing its toughness and increasing the microhardness and wear rate, whereas it did not exert a pronounced effect on dry friction coefficients. Sang et al. [23] showed that APS deposited NiCrBSi coatings from small diameter particles (50–75 μm) have diminished porosity but also smaller hardness values and corrosion resistance than coatings made from higher diameter particles (75–100 μm).

In this research, coatings made of AMDRY 1371 (Mo–NiCrFeBSiC) powder were deposited by the APS process in multiple successive passes on samples made of AISI 304 (EN 1.4301) steel substrate. The aim of this paper is to investigate the effect of coating thickness on its obtained microstructure and tribological properties, with no studies being published until now on thickness optimization for this coating.

## 2. Materials and Methods

### 2.1. Materials

The chemical composition of this powder, according to Oerlikon-Metco's online catalogue, is presented in Table 1. As can be observed, Mo is in a proportion of about 75 wt.%. The manufacturer recommends this powder with high molybdenum content for coatings with scuffing resistance, high toughness, and low friction coefficient, as well as its application for pump bushings and sleeves. The morphology of the powder consists of spheroidal particles with particle sizes of 90 + 25 μm. This blending powder has a melting point of 660 °C, service temperature ≤ 340 °C, as compulsory spray process being indicated APS or HVOF. The supplier indicated that the finishing of such coatings typically used the wet grind using a SiC or diamond wheel. In order to keep the morphology of the deposited coatings unmodified, no grinding was applied, and the coatings were studied as they resulted from the deposition process.

Parallelepiped samples of 100 mm × 10 mm × 5 mm were cut from a worn irrigation pump sleeve made of AISI 304 (EN 1.4301) [24]. Before the APS process, the steel samples were sandblasted and polished. Coated pads were realized by APS deposition of AMDRY 1371 (Mo–NiCrFeBSiC) on the steel substrate, using a Oerlikon-Metco 9MB gun (Pfäffikon, Switzerland) and SPRAYWIZARD

9MCE equipment (Sulzer & Metco, Pfäffikon, Switzerland), the technological parameters of the coating process are indicated in [24].

**Table 1.** Chemical composition of AMDRY 1371 (Mo–NiCrFeBSiC) powder.

| Product | Nominal Chemical Composition (wt.%) | | | | | | | |
|---|---|---|---|---|---|---|---|---|
| AMDRY 1371 | Mo<br>Balance | Ni<br>17.5 | Cr<br>4.0 | Fe<br>1.0 | B<br>0.85 | Si<br>1.0 | C<br>0.25 | Others<br><0.3 |
| AISI 304<br>(EN 1.4301) | C%<br>≤0.07 | Si%<br>≤1.0 | Mn%<br>≤2.0 | P%<br>≤0.045 | S%<br>≤0.015 | Cr%<br>17.5–19.5 | Ni%<br>8.0–10.5 | N%<br>≤0.11 |

Three samples were coated with 5, 7, and 9 successive deposition passes of AMDRY 1371, the samples being denoted in this paper as 5L, 7L, and 9L.

*2.2. Methods*

2.2.1. Surface Topography Measurement

The roughness of the tested samples and also the profile of the wear traces on samples were measured using a stylus profilometer (made by Taylor Hobson, Leicester, UK), model Form Talysurf 50 and the μltra Intra Form Talysurf interpretation software (Ultra Version 5.5.4.20). For the used standard stylus arm code 112/2009, the range/resolution was 1.0 mm/16 nm (0.04 in/0.64 μin).

2.2.2. Microhardness Measurements

The Rockwell microhardness of coatings was measured using the indentation modulus of CETR UMT-2 micro-tribometer (Luleå, Sweden), the microhardness mean values were obtained as the arithmetic mean of five tests. The flat coated samples were indented using a Rockwell diamond tip indenter (CETR, Luleå, Sweden), with an opening angle of $-120° \pm 0.35°$, radius $-200 \pm 10$ μm, deviation from profile $\pm 2$ μm. The Young's modulus, E, was also obtained. The indentation method consisted of a progressive increase in the indentation force from 0 to 5 N and then the return to the initial value. The capacitive sensor along with the force sensor allowed us to obtain the typical indentation diagram (force–deformation). The software for micro-indentation and the microscratch test was the CETR-UMT Test Viewer.

2.2.3. Morphological and Structural Analyses

Surface and cross-section SEM images of coatings were taken with SEM equipment Quanta 200 3D Dual Beam (Waltham, MA, USA). The EDS analysis was realized using the unit model XFlash (Bruker, Billerica, MA, USA). XRD analysis was carried out using an Expert PRO MPD facility from Panalytical (Almelo, The Netherlands), with a Cu X-ray tube (Kα-1.54051°) [25].

2.2.4. Tribological Friction and Wear Tests

The AMSLER machine, known also as a two-disc machine, was first fabricated by Alfred J. AMSLER & Co., Schaffhouse, Switzerland, and used for wear tests of metals under a wide variety of testing conditions [26]. Our AMSLER machine (Type A 135, made by Wolpert Werkstoffprüfmaschinen G.mb.H. in Schaffhausen, Switzerland) was used to test the tribological pairs composed of a fixed upper coated sample and lower rotating disc made of AISI 52100 rolling bearing steel with hardness 60–64 HRc, with an equal radius of 29.5 mm in both radial and axial directions, and a disc thickness of 10 mm. To keep the upper coated sample fixed, the upper gears transmission chain was interrupted [27]. A complete description of the testing machine is provided in [27–30]. For the sake of completeness, images of the AMSLER machine and the data acquisition system are also provided in this paper (Figure 1). Tribological tests were repeated thrice and mean values were reported. All the tests were carried out in dry friction conditions, with a constant applied load of 20 N, and constant speed

of 100 rpm, the running time of each test being one hour. The load was applied by dead weights. A data acquisition chain based on tensometric measurements and Vishay P3 strain gage bridge (made in Braunschweig, Germany) provided friction torque measurements. The acquired data were post-processed by a LabVIEW virtual instrument (Version: 7.1) [27–30]. The computation formulas for the mean friction moment, mean coefficient of friction (CoF), and the data acquisition chain calibration procedure are presented in [28].

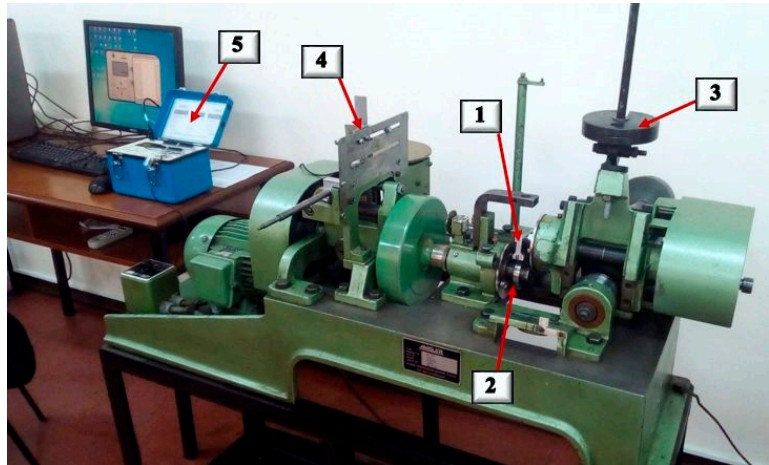

**Figure 1.** General view of AMSLER machine and data acquisition system.

In Figure 1, the next notations were adopted: 1—stationary coated sample; 2—AISI 52100 rotating disc; 3—dead weight loading system; 4—tensometric sensor; and 5—data acquisition system (type: Vishay P3 tensometric bridge with four channels, made by Vishay, in Braunschweig, Germany).

## 3. Results

### 3.1. Surface Topography

Before the tribological tests, the arithmetic mean roughness, Ra, of each tested sample surface was measured. Profilometry results are presented in Table 2. After each friction test, the rotating disc of AISI 52100 steel used in the friction and wear tests carried out on the AMSLER machine was polished, with the obtained mean roughness also being indicated in Table 2. The polishing was realized by using sandpaper grit 320. The aim of the polishing process was to obtain a similar roughness for all the tests but also to shorten the processing time of the surface. For the roughness line analysis, the next parameters were adopted: Gaussian filter, cutoff (Lc) 0.25 mm, cutoff (Ls) 0.008 mm, bandwidth 30:1.

**Table 2.** Roughness of the tested samples, $R_a$.

| Sample | 5L | 7L | 9L | Disc AISI 52100 |
|---|---|---|---|---|
| Longitudinal roughness, Ra [µm] | 8.228 ± 0.43 | 4.819 ± 0.24 | 6.741 ± 0.34 | 1.0 ± 0.1 |
| Transversal roughness, Ra [µm] | 9.236 ± 0.48 | 5.670 ± 0.29 | 6.030 ± 0.27 | 1.2 ± 0.1 |

The variation of the roughness of the obtained samples is due to the nature of the APS process, the successive passes erratically conducted to splat-on-splat melted powder structures.

### 3.2. Hardness and Elasticity Modulus

Rockwell microhardness HR, in GPa, was measured. As the adopted test was HR0.5 (preload 0.5 N and maximum applied load of 5 N), the indenter maximum displacement was below 12 µm,

far from the thickness of the deposited coatings (over 50 μm). For this reason, global values of the microhardness and elastic modulus were computed as mean values of five measurements (Figure 2). It can be seen that the reduced indentation modulus, E0, is near to Young's elasticity modulus (Table 3).

**Table 3.** Rockwell microhardness HR0.5, and Young's elasticity modulus, E.

| Parameter | AMDRY 1371 | Confidence Limits |
|---|---|---|
| Microhardness, $HR_{0.5}$ [GPa] | 0.464 | −31.21% + 34.32% |
| Young modulus, E [GPa] | 64 | −15.03% + 18.14% |
| Reduced modulus, E0 [GPa] | 65.5 | −14.28% + 16.94% |

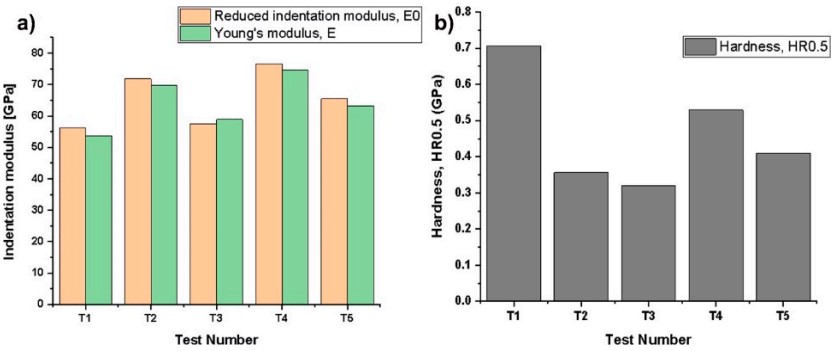

**Figure 2.** Indentation modulus (**a**), and HR0.5 microhardness (**b**).

As the EDS results proved (see Table 4), the composition of the coating is different from point to point. Additionally, the surface roughness may influence the indentation results. Regarding the microhardness, it seems that it depends on the APS spraying distance [31], which was kept constant in our research. As we observed, the measured microhardness of coatings depends on the indentation point, the spread of the results being acceptable for indentations at micro level. A close value of Young's modulus, E = 55 ± 6 GPa, was reported for APS deposited AMDRY 1371 in [32].

*3.3. Morphological Analysis of Unworn Coatings*

3.3.1. SEM Analysis

Images of surface SEM analysis of the base material and coated samples are presented in Figure 3. The base material microstructure is composed of evenly distributed α-ferrite (darker) and perlite (lighter) grains (Figure 3a). The AMDRY1371 powder coated surface possesses a morphology characteristic to the APS deposition process, with splats, pores, and rare partially melted powder particles (see Figure 3b,c). It must be noticed that the semi-melted light spherical particles are of Mo (Figure 3c), as proved by these EDS results and confirmed by [19]. At high magnification of 5000× (Figure 3d), the increased molybdenum content (over 75%) formed a flake-like microstructure, which may protect the surface during the dry running-in period of the tribological tests.

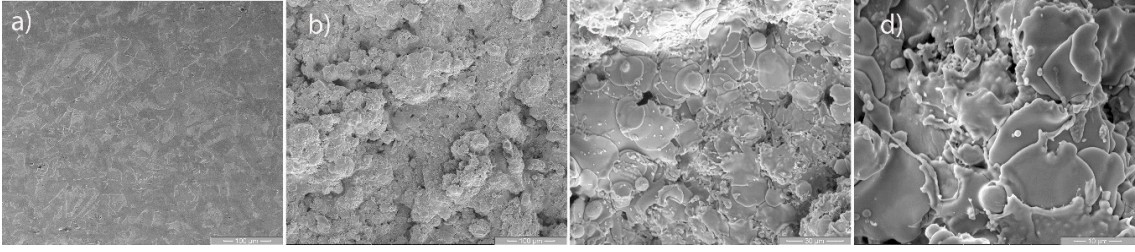

**Figure 3.** Surface SEM images of base material (500×) (**a**), and AMDRY 1371 coated sample (**b**) 500×; (**c**) 2000×; (**d**) 5000×.

Figure 4 contains the results of the cross-section SEM analysis, transversal to surface coating, for all the samples, at various magnifications. The left side of Figure 4 presents measurements of the thickness of coatings, with the mean thickness values obtained from six equidistant measurements being: 5L = 58.8 μm, 7L = 82.5 μm, and 9L = 100.0 μm.

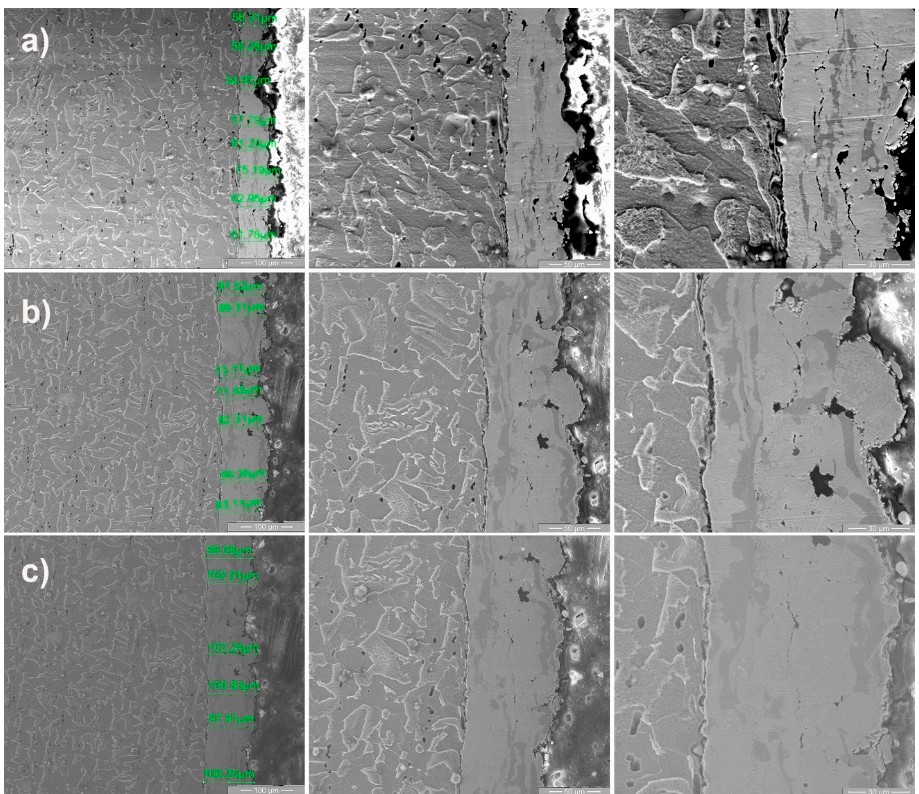

**Figure 4.** Cross-section SEM images of 5L (**a**), 7L (**b**), and 9L (**c**) coatings at various magnifications, from left to right: 500×, 1000×, and 2000×.

The thickness of the deposited coatings is much less than reported by [19], which is about 300–400 μm. As the present friction and wear results will prove, a very large thickness is not always beneficial from the viewpoint of the dry wear rate of sliding contacts. Some pores and micro cracks can be observed at a higher magnification of 2000×. As seen from the SEM images, the microstructure of these coatings is better than that obtained by Zhang et al. [20] for 5 wt.% to 30 wt.% added Mo in NiCrBSi, with less pores and voids observed (Figure 4).

### 3.3.2. EDS Analysis

The EDS results for the unworn surfaces of coated samples were obtained as mean values (wt.%) of four measurements at random points of the coatings (Figure 5). Additionally, computed global mean values of elements (wt.%) for all the AMDRY 1371 samples are presented in Table 4.

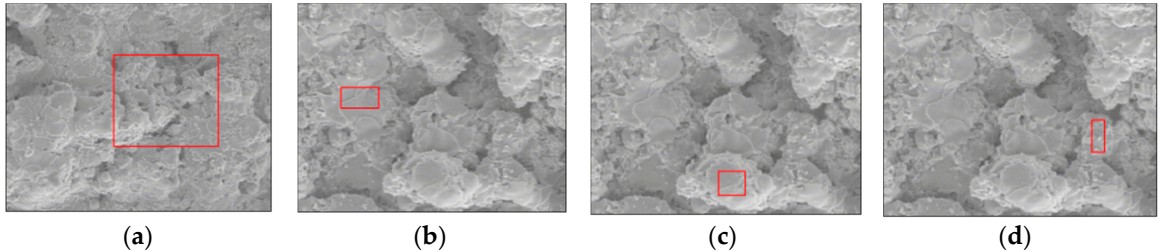

| (**a**) | (**b**) | (**c**) | (**d**) |

**Figure 5.** EDS over AMDRY 1371 coating surface (**a**) test 1; (**b**) test 2; (**c**) test 3; and (**d**) test 4.

**Table 4.** EDS results for the surfaces of coated samples after the atmospheric plasma spray (APS) deposition process, before friction tests.

| Elements wt.% | Samples | | | | |
|---|---|---|---|---|---|
| | Mean Values (Test 1–Test 4) | Test 1 | Test 2 | Test 3 | Test 4 |
| Mo | 79.248 | 77.57 | 80.33 | 81.62 | 77.47 |
| Ni | 5.605 | 6.37 | 3.15 | 5.63 | 7.27 |
| Cr | 1.323 | 1.29 | 0.93 | 1.31 | 1.76 |
| Fe | 0.825 | 0.67 | 0.96 | 0.99 | 0.68 |
| O | 6.267 | 7.38 | 7.64 | 3.49 | 6.56 |
| Si | 0.318 | 0.33 | 0.25 | 0.20 | 0.49 |
| B | 6.418 | 6.40 | 6.74 | 6.76 | 5.77 |

Comparing the composition of the commercial deposited powder (Table 1) against the mean values from the surface EDS analysis of the deposited coatings (Table 4), it seems that Mo has the tendency to migrate to the surface during the APS deposition process, while Cr remains in the substrate in a larger quantity. The presence of Mo to the surface zones is a gain from the viewpoint of friction reduction capabilities.

In Figure 5, we emphasized the EDS analyzed surfaces and we reported mean values in Table 4. For pertinent mean values of elemental analysis, in Figure 5a we selected a higher area containing some voids. The idea was to compare the obtained general mean values, including the analysis of selected specific smaller different areas, with the results of this measurement influenced by the existence of these voids. As observed, the general mean values are between the results of test 1 and the mean of tests 2 to 4, as it should be for a porous structure.

Figure 6 presents the results of line scan EDS analysis in cross-sections for all 5L, 7L, and 9L samples. The base material is observed on the left side of each subpicture. The increase in Mo content over the thickness of deposited coatings is accompanied by a strong decrease in Fe content for all the samples.

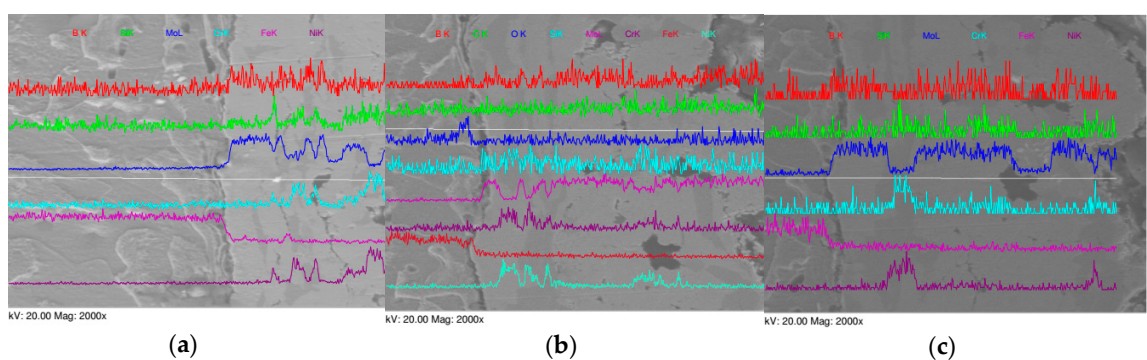

(**a**)  (**b**)  (**c**)

**Figure 6.** EDS line analysis over cross-sections for (**a**) 5L; (**b**) 7L; (**c**) 9L (for details, see Supplementary Material, Figures S1–S3).

### 3.4. Friction Tests on AMSLER Machine

Friction tests were carried out on the AMSLER machine in dry conditions, at constant load conditions and constant speed. The friction moment evolution in time was monitored by a data acquisition system [27,28], and the mean values of the friction coefficients were obtained by post-processing of acquired data using a developed LabVIEW virtual instrument [27].

At the beginning of the tests (Figure 7a), the friction torque of sample 5L was less than that of 7L and 9L, but it undertook an ascendant trend as the thinner coating was removed by abrasion. Its dynamic evolution in the range of 1000–2000 s is supposed to be due to the direct contact between the AISI 52100 surface of the counterpart test roller and the base material surface. Thicker coatings

presented a constant linear evolution of the friction torque (9L), and descendent evolution for sample 7L. The lowest coefficient of friction (CoF) was obtained for sample 5L, but the CoF values increased slowly for the thicker coatings—7L and 9L. The evolution of the 7L friction coefficient was as generally expected, with a typical curve being indicated in [33]. Regarding the obtained values of CoF, around 0.3, they are better than those reported by Dilawary et al. [21], who found values of CoF increasing from 0.4 to 0.9, for tests at room temperature on PTA deposited Mo(10%)-NiCrBSi and a sliding distance of 500 m.

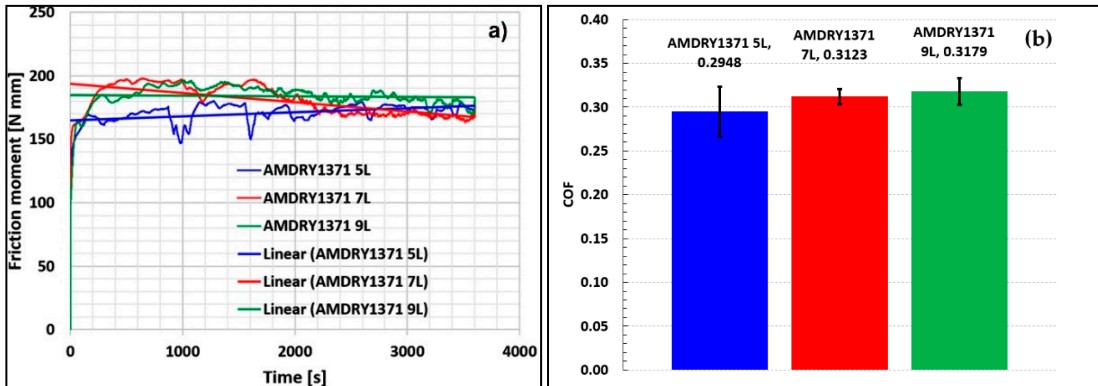

**Figure 7.** (**a**) Friction moment versus time, and (**b**) mean values of coefficient of friction.

The fluctuation of the friction moment during tests affected the mean values of the friction coefficients (see Figure 7b). To interpret the data, friction results must be correlated with wear results, EDS, and XRD analysis, with this aspect being treated later in the Discussion section.

### 3.5. Wear Analysis of Coatings

The wear analysis of the coatings includes SEM analysis of the wear spots and their vicinity (Figure 8), EDS analysis of worn coatings, and XRD analysis.

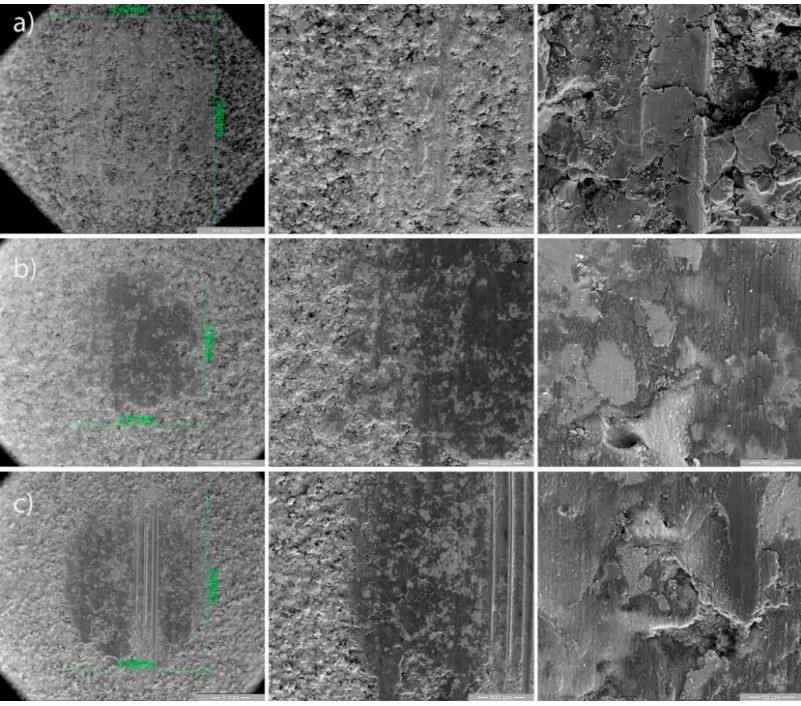

**Figure 8.** SEM images of wear traces of 5L (**a**), 7L (**b**), and 9L (**c**) coatings at various magnifications, from left to right: 50× (overall contact), 100× (border of worn–unworn), and 1000× (wear spot).

### 3.5.1. SEM Analysis

The left side of Figure 8 contains magnification at 50× of the wear spots of each coating with dimension lines of each wear scar: 5L (Figure 8a), 7L (Figure 8b), and 9L (Figure 8c). The SEM images of sample 5L at higher magnification (Figure 8a—1000×) expose crashed material and micro cracks, typical of combined abrasive and adhesive wear. It can be seen that 7L has the smallest wear scar area, while 5L has the biggest worn area. The images from the middle column of Figure 8 present the border area between the worn and unworn coated zones of each sample, but at higher magnification (100×). Samples 7L and 9L reveal a glassy microstructure (darker area), indicating the presence of Si and $MoO_2$ oxides and intensified sliding friction between the glossy contact surfaces. Scratches can be seen in the middle of the wear spot of sample 9L, indicating abrasive scratching with hard particles. As the XRD analysis has shown, there are some CrB particles on the wear scar of the 9L coating. An agglomeration of such particles at the bottom of the wear cup, entrapped in the Mo substrate, adhered to the similar metallic component from the counterpart surface of the AISI 52100 testing roller and produced these scratches.

### 3.5.2. EDS Point and Line Analysis of Worn Coatings

To identify light and dark regions within the wear spots, EDS analysis was also conducted according to Figure 9 and Table 5. From the results, it can be observed that the lighter area is rich in Mo (59.38 wt.%) and O (19.21 wt.%), being composed of pure Mo and $MoO_2$, as the XRD revealed. The darker area is mainly composed of iron, Fe 64.15 wt.%, and $FeNi_3$—as proved by XRD analysis. A small proportion of O is found in the dark zone (2.58 wt.%).

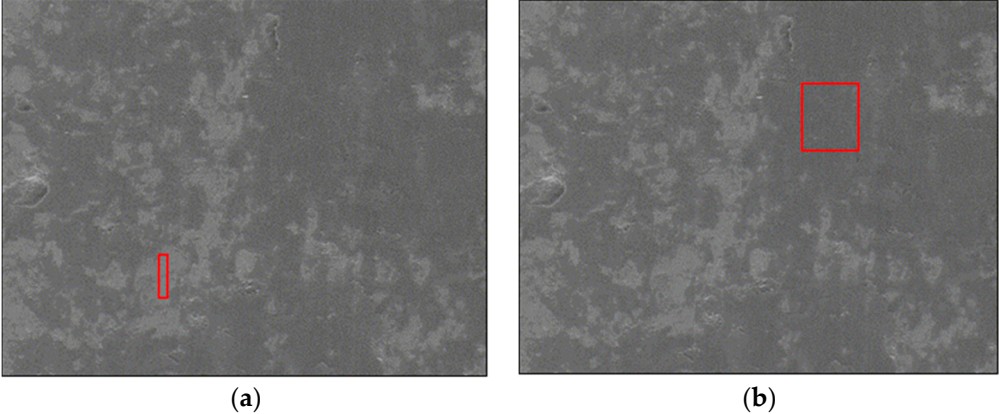

|       | (a)       |          | (b)       |
|-------|-----------|----------|-----------|

**Figure 9.** EDS of light zone of wear scar (**a**), and dark zone of wear scar (**b**) at 500×.

**Table 5.** EDS results on light and dark zones of wear scar of sample 7L.

| Elements wt.% | Sample 7L | |
|---|---|---|
| | **Light Zone** | **Dark Zone** |
| Mo | 59.38 | 6.89 |
| Ni | 4.66 | 1.69 |
| Cr | 2.15 | 1.68 |
| Fe | 14.31 | 64.15 |
| O | 19.21 | 2.58 |
| Si | 0.29 | 0.5 |

Figure 10 presents the results of line scan EDS analysis over the worn area of all the tested samples. It can be seen that the scan line over each sample started from the unworn coating (left side), and passed over the boarder unworn to worn surfaces. In such a way, the fluctuation of each element wt.% over the scanning line was emphasized.

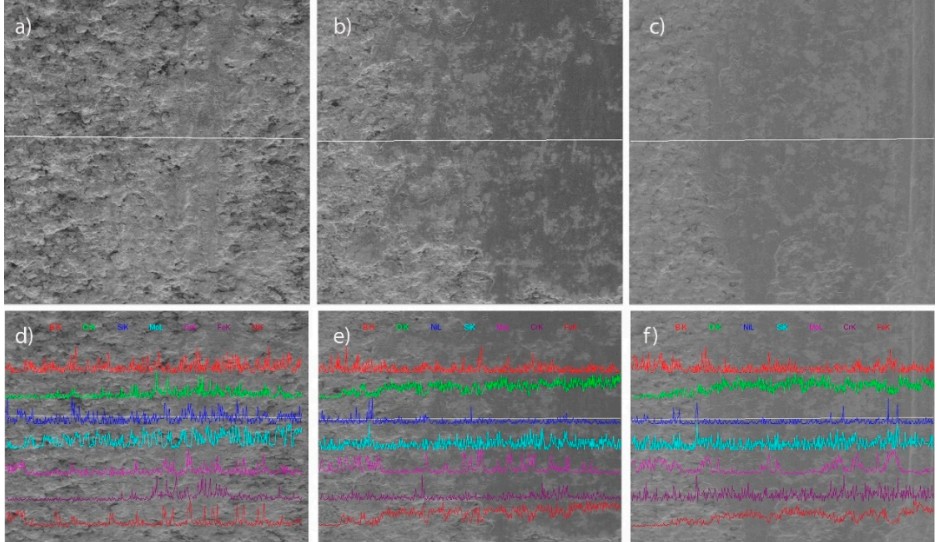

**Figure 10.** EDS line analysis over the worn surfaces: SEM images (50×) of EDS line direction for (**a**) 5L; (**b**) 7L; (**c**) 9L, and EDS line analysis for (**d**) 5L; (**e**) 7L; (**f**) 9L (for colors see Supplementary Material, Figures S4–S6).

The effects of the various elements on the friction and wear of each sample will be treated later, in the Discussion section.

### 3.5.3. XRD Analysis of Coatings

A 3D map of elemental distribution over the base material, with coatings without wear and worn 5L, 7L, and 9L samples, is provided in Figure 11. The highest peaks correspond to Mo, and in their right and left near vicinity there are peaks associated with $MoO_2$. The obtained distribution of the elements is similar to [19], a reference that also studied Mo-NiCrFeBSi (75–25 wt.%).

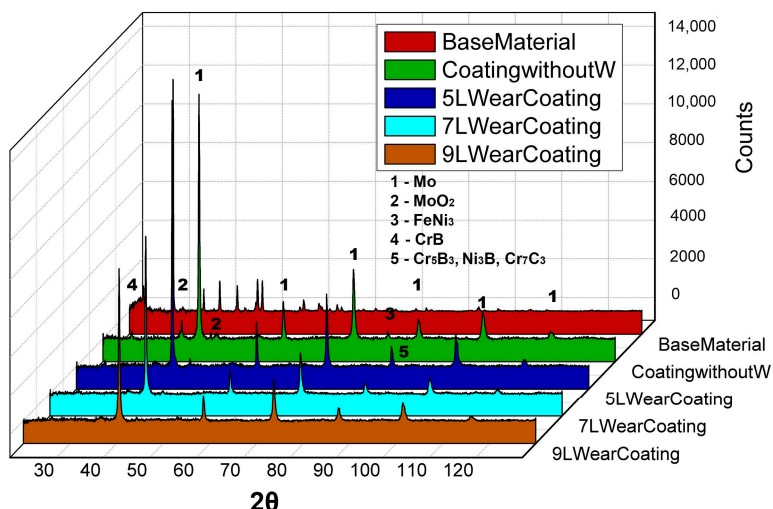

**Figure 11.** XRD analysis of base material, and deposited coating surface.

### 3.5.4. Wear Profiles and Wear Rates

In order to obtain the exact wear volume of each coated sample, the measurements of the worn area from SEM images (Figure 8) were combined with the information from worn profiles (Figure 12).

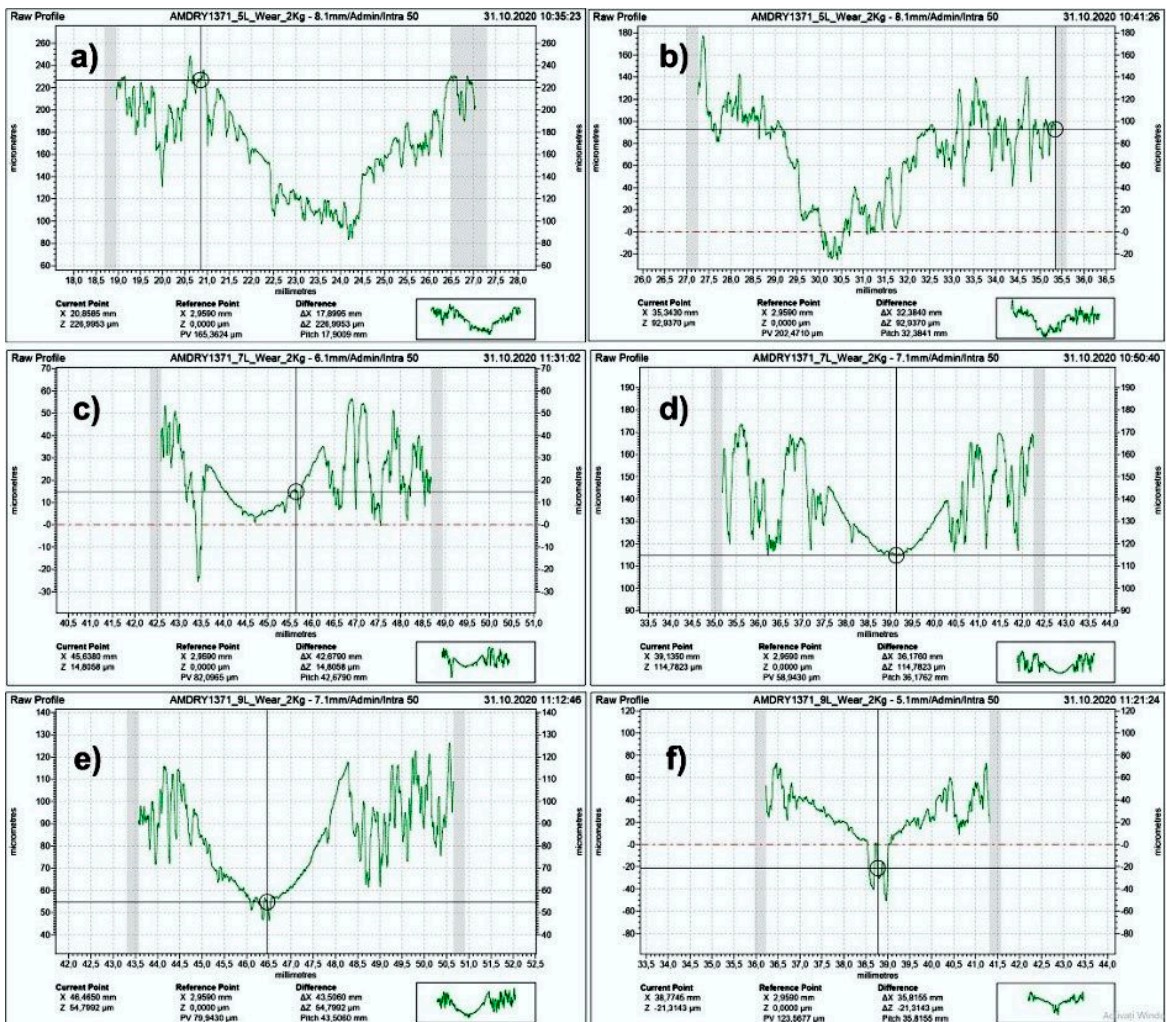

**Figure 12.** Wear profiles of coatings in longitudinal direction (**a**) 5L, (**c**) 7L, and (**e**) 9L, and in transversal direction (**b**) 5L, (**d**) 7L, and (**f**) 9L.

The deep scratches in the middle area of sample 9L can be seen by the transversal profile image from Figure 12f, this remark being in close correlation with the results presented in Section 3.5.1. The computed wear rate, *W*, in mm³/N·m, expressed by Equation (1), from [33], is represented in Figure 10.

$$W = V/(Q \cdot L) \tag{1}$$

$$V = 2 \cdot a \cdot b \cdot c/3 \tag{2}$$

$$L = \pi \cdot D \cdot N \cdot T/1000 \tag{3}$$

where *V* is volume of wear of the coated sample in Equation (2), computed as the volume of the demi-ellipsoid with the semiaxes *a*, *b* and *c* measured from raw wear profiles, *Q* is the applied normal load on the contact (20 N), *L* is the total sliding distance given by Equation (3), in this case 1112 m. The AISI 52100 roller diameter is 59 mm, the speed *N* = 100 rpm, and the time *T* = 3600 s.

The highest wear volume and wear rate were obtained for sample 5L, while the lowest corresponded to sample 7L (Figure 13). An explanation for the increased wear volume of 5L is provided by the existence of a higher proportion of hard compounds based on Cr, B, and C; this being denoted as peak five from Figure 11. The error bars are in close correlation with the variation of the measured friction torque, the influence of friction on the wear process being evident.

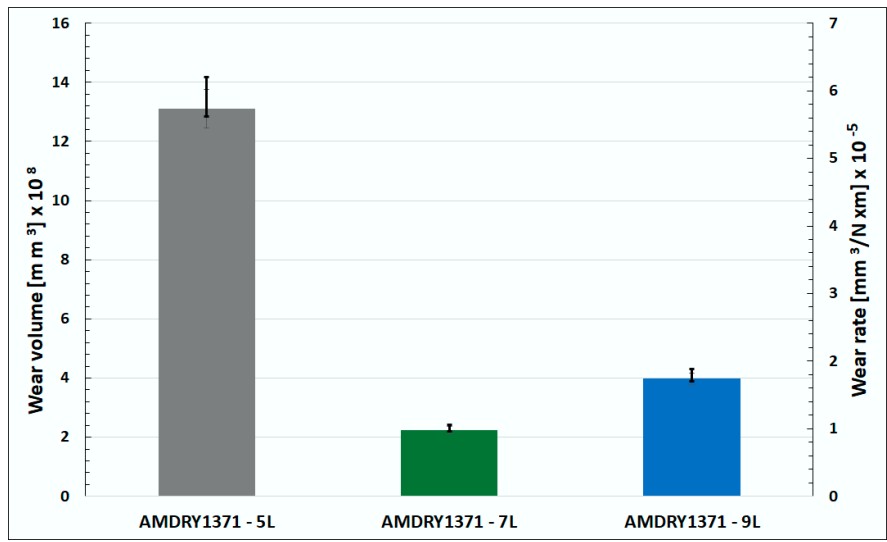

**Figure 13.** Wear volume (**left axis**) and wear rates (**right axis**) of tested coated samples.

## 4. Discussion

In reference to previously published research on similar subjects, only Niranatlumpong and Koiprasert [19] have studied both the microstructure and the tribological properties of APS coatings from Mo–NiCrFeBSi (75–25 wt.%) powder. The results reported in [19] are not positive for the Mo–NiCrFeBSi (75–25 wt.%) coating, that is AMDRY1371. In tribological tests of [19], the tested materials were the same as in this paper, and even the applied load was similar: 25 N in [19], and 20 N in this research. Dry conditions and pure sliding were chosen in both tests, but the most important aspect is the existing Hertz pressure, $\sigma_H$, between the contact bodies, computed according to Equation (4).

$$\sigma_H = 3 \cdot Q/(2 \cdot \pi \cdot a \cdot b) \tag{4}$$

where $Q$ is the applied load on the contact, $a$ and $b$ are the semimajor and semiminor axes of the elastic elliptical contact [34]. In fact, [19] warned that this coating is not suited for very high pressures, but a high pressure was obtained by choosing a small diameter ball, even if a reduced load was applied. The ball with a 6.3 mm diameter used as counter-body in [19] creates a Hertz pressure of 1135 MPa, corresponding to high loads and severe exploitation regime, typical for mechanisms with concentrated contacts (e.g., rolling bearings and gears), while the computed Hertz pressure in our contact, created by a disc of 59 mm diameter, was only 237 MPa, as encountered in highly loaded sleeve bearings, this being the aim of this study. This aspect, correlated with the thicker coatings (350–400 μm) deposited in [19], led to the rapid crash of the coating containing 75% Mo and 25% NiCrFeBSi, even if the testing time was reduced (1000 s). As Marquer et al. [4] reported, the frictional properties are more dependent on the contact pressure rather than on the sliding velocity.

Regarding the results on the friction coefficient during our tribological tests (Figure 7), they are closely related to the different roughness of the tested samples (Table 2). This supposition is confirmed by the findings of [19], where reduced friction coefficients of 0.1–0.15 were obtained by polishing the coatings up to $R_a$ = 0.5 μm. The 5L samples had a higher roughness, from here the primordially flake-like Mo particles of the asperities (see Figure 3) were broken from the beginning and entered the contact as solid lubricant, assuring a mild abrasive wear and extended contact area. The evolution of the friction coefficient fluctuated after 1000 s, the time supposed to be necessary to remove the thinner coating of sample 5L. After that, the metal-to-metal contact of the base material and the testing steel counter-body contributed to a continuous increase in the friction torque. However, the EDS and XRD analysis proved the existence of a high quantity of Mo in the contact area of sample 5L at the end of the test, demonstrating the existence of solid lubrication regime over an extended contact

area with diminished real contact pressure, the obtained mean friction coefficient being the lowest. The results of XRD analysis (Figure 11) confirmed the findings of Dilawary et al. [21], which indicated the presence of $Cr_5B_3$, $Ni_3B$, and $Cr_7C_3$ around 83 degrees (peak five, Figure 11). They asserted that Mo refined the microstructure of hardfacing NiCrBSi, introducing a new $Mo_2(B, C)$ type boro-carbide phase. All these compounds containing Cr and B are hard particles, and they were found on the map of Figure 11 in a greater proportion on the worn surface corresponding to sample 5L. This explains the accelerated abrasive wear of 5L, the phenomenon being accompanied by micro adhesions between similar metallic compounds of contacting bodies (see Figure 8a, 1000×). Furthermore, as the line scan EDS analysis shows, the 5L wear scar contains an increased percent of iron just over the severe worn area (Figure 10d, Fe is magenta color). The quantity of Mo (see XRD results, Figure 11), O, and Ni in the worn area is also abundant, from here the reduced friction coefficient, which had an ascendant evolution due to the contact between the bare base material surface and counterpart steel roller surface, amplified over the time. In comparison, more O was found over the worn area of samples 7L and 9L (Figure 10e,f) and the Mo and $MoO_2$ compounds protected the contact area and slowed down the wear rate. On the scratched area of sample 9L (Figure 8c), there is an increased amount of Fe, Cr and B (Figure 10f), that is, agglomeration of hard particles, but a reduced quantity of Mo. High Mo content is found near these scratches, confirming that the agglomeration of hard particles of Fe, Cr and CrB was entrapped in the Mo, producing the scratching in the sliding motion direction, due to the high applied pressure on sliding contact. The highest distribution of Mo was found in the center of the contact area of sample 7L, explaining the reduced wear. The positive result is confirmed by Dilawary et al. [21], who emphasized the beneficial contribution of 10% added Mo to NiCrBSi powder, with the wear resistance increasing two-fold.

Regarding the wear modes, specific to all the coatings deposited by APS, the SEM images confirmed splats' delamination and abrasion. Furthermore, by varying the thickness of the coatings, it seems that the thin 5L coating was subjected to both abrasive and adhesive wear over time, while the thicker coating favorized agglomeration of Fe, Cr, B hard particles and the initiation of abrasive ploughing wear. The surface of sample 7L had no scratches and became glassy at the end of the friction test, indicating a mild abrasive wear at the interface surface of the coating and a high content of O and Mo being detected by EDS line analysis of the worn surface.

Comparing the obtained wear rates with results from the literature, the 7L and 9L samples displayed better behavior than HVOF sprayed Colferoloy–NiCrFeSiBC coatings ($1 \times 10^{-4}$ mm$^3$/N·m) tested at room temperature [11]. For a similar coating, [19] only reported results on wear depth (175 µm), whereas our samples presented maximum depths of the wear path around 95, 50, and 55 µm for samples 5L, 7L, and 9L, respectively, and even the testing time in [19] was only 1000 s and the initial roughness of samples was diminished by polishing ($R_a = 0.5$ µm).

## 5. Conclusions

Mechanical components of irrigation and slurry pumps experience various kinds of wear during exploitation. In order to combat wear, these components can be coated with diverse types of wear resistant coatings. AMDRY 1371 (Mo–NiCrFeBSiC) is a high molybdenum content (75 wt.%) coating, Oerlikon-Metco's online catalogue recommends it as a coating for pump bushings and sleeves.

In this research, coatings made of AMDRY 1371 (Mo–NiCrFeBSiC) powder were deposited by the APS process in multiple successive passes on parallelepipedal AISI 304 (EN 1.4301) steel samples manufactured from a worn sleeve of a multistage vertical irrigation pump. The aim was to investigate the effect of coating thickness on its obtained microstructure and tribological properties. To find an optimum thickness of AMDRY 1371 coating, the samples were coated by five, seven and nine passes (counted as return passes of APS gun), and denoted as 5L, 7L, and 9L, respectively.

Mechanical properties of the coating (microhardness $HR_{0.5}$ and Young's modulus) were determined by micro-indentation tests. $HR_{0.5}$ microhardness and Young's modulus were assessed as mean values

of five indentation tests, with $HR_{0.5} = 0.464$ GPa and $E = 64$ GPa. A close value $E = 55 \pm 6$ GPa was reported for the same coating in [32].

Friction behavior, wear resistance, and wear modes of the coated samples were investigated by tests on AMSLER tribometer at a constant load (20 N), constant speed (100 rpm), and dry conditions, the span of each test being one hour. The obtained mean value of CoF was found to be around 0.3 for all the samples. Depending on the thickness of the coatings, initial roughness, and their microstructure, the evolution of the friction moment during the one hour of the test and also the final wear volumes and wear rates were totally different.

Combining the information gathered from all the tests, it was found that high roughness and a thin coating attracts metal-to-metal contact between the surfaces of the base material and counterpart steel roller used in the tribological tests, and after about 1000 s the thin coating of sample 5L was partially removed, and an accelerated wear process was started. The lowest wear volume and wear rate was obtained for the 7L coating, and for the thicker coating 9L, scratches were found on the wear surfaces, due to the agglomeration of Fe, Cr, and B hard particles.

Comparing the obtained wear rates with results from the literature, samples 7L and 9L proved similar or better results. According to our results, there is an optimum value of the coating thickness for each application, and improvements to the friction and wear properties are possible by decreasing the surface roughness of the coating by wet polishing, as recommended by the powder manufacturer.

The reported results have both applicable and cognitive characteristics. For the short term, such a coating can be applied to the investigated mechanical component of irrigation pumps that are subjected to abrasive wear. In addition, our findings have a cognitive characteristic too, as similar results have been previously reported for a different coating [30].

Future research on the subject should aim tests towards oil lubricated conditions at various loads and speed. Additionally, friction and wear results will be correlated with theoretical results on developed equivalent stress in coatings and substrates, as the thickness of the coating may influence the position point of the maximum stress relative to the coating and base material interface.

**Supplementary Materials:** The following are available online at http://www.mdpi.com/2079-6412/10/12/1186/s1, Figure S1: Line scan EDS results—Cross section—5L sample -AMDRY1371, Figure S2: Line scan EDS results—Cross section—7L sample -AMDRY1371, Figure S3: Line scan EDS results—Cross section—9L sample -AMDRY1371, Figure S4: Line Scan EDS results over worn surface—5L sample—AMDRY1371, Figure S5: Line Scan EDS results over worn surface—7L sample—AMDRY1371, Figure S6: Line Scan EDS results over worn surface—9L sample—AMDRY1371.

**Author Contributions:** Conceptualization, C.M., C.C.P., P.V., B.I., and V.P.; methodology, C.C.P., C.M., P.V., B.I., S.B., and V.P.; software, V.P., B.I., S.B., and M.S.B.; validation, V.P., B.I., and S.B.; formal analysis, P.V., C.M., S.B., V.P., B.I., M.S.B., and C.C.P.; investigation, B.I., V.P., C.C.P., and S.B.; resources, C.M. and P.V.; data curation, C.M., B.I., and V.P.; writing—original draft preparation, C.C.P., V.P., B.I., and S.B.; writing—review and editing, C.M., P.V., V.P., B.I., and S.B.; visualization, B.I., V.P., and C.C.P.; supervision, C.M. and P.V.; project administration, C.M.; funding acquisition, C.M. and P.V. All authors have read and agreed to the published version of the manuscript.

**Funding:** This research received no external funding.

**Conflicts of Interest:** The authors declare no conflict of interest.

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
