# Peer review of "Microstructural Analysis and Tribological Behavior of AMDRY 1371 (Mo–NiCrFeBSiC) Atmospheric Plasma Spray Deposited Thin Coatings"

_coatings, doi:10.3390/coatings10121186_

Round 1

Reviewer 1 Report

This is an interesting area of research, however, there is a need for improvement.

The weaker side of the study is the very narrow range of the research. Only one load, only one speed and only one period of time. I’m afraid it is too small to obtain wider conclusions in that area of research. Other remarks are as follows:

  1. p. 2.1. Authors mentioned that before the APS process, the steel samples were sandblasted and polished. What devices and treatment parameters were used for these processes?
  2. p. 2.2.4. How many times tribological tests were repeated?
  3. p. 3.1. Are you sure that value of Ra parameter after polishing process presented in Table 2 is arround 1 µm? Such values you can obtain by turning or milling. After polishing values close to 0.1 µm or even smaller are expected.
  4. p. 3.4. in Figure 7b error bars or scatters should be marked. The same situation with Figure 13. Depending on the presented values (error bars or scatters) some comments should be provided.
  5. p. 5. The subsection conclusions should be modified. Such sentences like „The roughness of the samples and the wear volumes were measured using a Taylor Hobson profilometer. To understand the nature and evolution of wear in various thickness coatings, the unworn and worn surfaces of the coated samples were analyzed by Scanning Electron Microscopy (SEM), Energy-Dispersive X-ray spectroscopy (EDS), and X-Ray Diffraction (XRD)” and others are not conlusions.
  6. It would be also worth clarifying where the results could be applied. Do the results have only cognitive or also applicable character?

Author Response

We are grateful to the all the reviewers for their priceless comments and suggestions which have helped us to enhance the quality of our paper. We have considered all your suggestions, the updated text of the article being highlighted in the revised manuscript.

Reviewer 2 Report

AMDRY 1371 coatings were deposited by Atmospheric Plasma Spray (APS) method on the sample to investigate the mechanical properties, the wear resistance and other surface characteristics. Overall, the research is well written and the experimental design is appropriate. However, some explanations and modifications are needed:

1. In the abstract, line 29. Taylor Hobson profilometer should be changed. the readers do not know what it is. The authors should use Stylus profilometer or else to show the instrument you use.

2. Abstract line 34, the authors should specify what application, rather than each application

3. Introduction: the authors should give some review or introduction on atmospheric plasma spray method, and surface characterization method. For example,

Azarmi, F., Coyle, T. W., & Mostaghimi, J. (2008). Optimization of atmospheric plasma spray process parameters using a design of experiment for alloy 625 coatings. Journal of Thermal Spray Technology, 17(1), 144-155.

Rico, A., Rodriguez, J., Otero, E., Zeng, P., & Rainforth, W. M. (2009). Wear behaviour of nanostructured alumina–titania coatings deposited by atmospheric plasma spray. Wear, 267(5-8), 1191-1197.

 Characterization of fused silica surface topography in capacitively coupled atmospheric pressure plasma processing

4. In 2.2.1, the authors should illustrate what is the precision and related specificationso of the instrument. And why not use optical measurement such as white light interferometer and chromatic confocal?

5. In 3.1, the authors should mention the pre-processing step and filter parameters. The Ra should be given with these conditions. The authors can refer to

Blunt, L., & Jiang, X. (2003). Advanced techniques for assessment surface topography: development of a basis for 3D surface texture standards" surfstand". Elsevier.

Li, D., Qiao, Z., Walton, K., Liu, Y., Xue, J., Wang, B., & Jiang, X. (2018). Theoretical and experimental investigation of surface topography generation in slow tool servo ultra-precision machining of freeform surfaces. Materials, 11(12), 2566.

6. In 3.3.2, why not use XPS to measure elements?

Why the region of measurement is different in Fig5

Author Response

(The authors gave the same response as above.)

Reviewer 3 Report

The manuscript ‘Microstructural analysis and tribological behavior of AMDRY 1371 (Mo-NiCrFeBSiC) atmospheric plasma spray deposited thin coatings’ has discussed in detail about the AMDRY coatings, its characterization and wear properties. It is written with many literature citations and it can be accepted after a minor revision based on the below comments.

  1. Line 48: Molybdenum Dialkyl Dithio Carbamate (MoTDC) or MoDDC?
  2. There are many subscripts to be corrected throughout the text
  3. Line 55: what is the role of SiC by using with Ni-60 since SiC was mainly discussed.
  4. Line 63; wear enhancement?
  5. Line 89 and 116: what are the differences or attempts being made in this study as compared to the said paper?
  6. Line 125: Can you provide the FESEM image of the powder?
  7. Line 137: After every passes, whether the coating was dried or any other conditions used?.
  8. Name the parts mentioned in Figure 1a.
  9. There were changes in the mean roughness values mentioned in table 2 after 5L,7L and 9L, what would be the reason?.
  10. Line 203: Images of
  11. Line 222,224: modify the word …our… into present work or this work…
  12. Line 223: The presence of Mo in AMDRY is much, thus it is quite natural to observe more quantities in EDS. Thus explanation of migration… does not require. Sentence needs modification or reference.
  13. Line 280: There is no fig. 9f, check it.
  14. 11; Is only the peaks for Mo detected, not other elemental diffractions?
  15. Line 332; name the investigators (eg. Yy et al) of [17].
  16. Line 397: exploitation of

Author Response

(The authors gave the same response as above.)

Round 2

Reviewer 1 Report

I can accept the manuscript in this form.

Reviewer 2 Report

I am fine with the corrections.